# OpenReview forum: "GASP: Efficient Black-Box Generation of Adversarial Suffixes for Jailbreaking LLMs"
_ICLR.cc/2025/Workshop/BuildingTrust — BuildingTrust_

### Official Review · Reviewer_eyVt · 2025-02-23
**Accept**

**Rating:** 8
**Confidence:** 2

**Review:**

The paper introduces a new blackbox (i.e. does not require access to the model gradients or any internals) method for generating adversarial suffixes for jailbreaking LLMs, which according to their experiments outperforms all existing methods.

The key (original) components of the design for finding “optimal adversarial prompts” seem to be:

- A *constrained* optimization, to maximize the probability of an adverserial response while adhering to the constraint that the prompt is natural based on some distribution $p_{\mathrm{nat}}$ that measures the probability that some text is “natural”.
- Optimization is done through the embedding space via “Latent Bayesian Optimization” (as opposed to e.g. greedy co-ordinate gradient style techniques that optimize on the token space directly)
- This optimization trains the parameters of a certain `SuffixLLM` which produces the optimal suffixes to adversarial prompts to elicit “harmful responses” — looking at Appendix I, this is done via ORPO.
- Evaluation of responses is done via another LLM `JudgeLLM`, as part of the `GASPEval` procedure, which produces the training signal for `SuffixLLM`.

Some things I didn’t understand or am not sure about:
- Re: the AdvSuffixes dataset — if I understand correctly, these are suffixes which adhere to the naturalness constraint, but are not actually successful at jailbreaking the LLM? Then what’s the point of using these suffixes rather than any random ones — is the idea that the probability of an adversarial response is higher for AdvSuffixes than for some random prompts?
- The paper claims their approach is totally blackbox and doesn’t even depend on the logits of the target LLM. But there is constant reference to the target LLM as a probabilistic model i.e. $p(y\mid x)$ etc. — are these not the (exponential of the) logits? How else do you model the probability distribution?

Perhaps these need to be explained better, or perhaps I’m just being dumb.

Regardless, the paper represents a valuable contribution to the field and is very relevant to the workshop topic. It should be accepted.

---

### Official Review · Reviewer_2XnX · 2025-02-28
**Accept, novel & impactful but with some key details inaccessible**

**Rating:** 8
**Confidence:** 2

**Review:**

This paper introduces GASP, a framework for generating adversarial suffixes that, when appended to harmful prompts, cause a target black-box LLM to produce harmful outputs. To improve efficiency over existing methods, GASP optimizes directly in the continuous embedding space rather than over discrete token sequences. It also incorporates coherence constraints to ensure that the adversarial suffixes are natural-sounding.

The discussion on the shortcomings of existing jailbreaking approaches is convincing -- they effectively outline the efficiency constraints of manually designed adversarial prompts vs. the incoherence and computational cost of optimization-based methods.

Their method consists of two main steps. The first, pretraining their SuffixLLM, is well-detailed and clearly explained, with a well-defined objective function. The second step—alignment and refinement using LBO and ORPO—shows promise but lacks crucial details in the main text. For instance, a brief explanation of GASPEval (which is detailed in the Appendix) would greatly improve comprehension of the LBO and ORPO refinement steps. Here, another clarification would be helpful: how is $p_{nat}(.)$ determined -- based on which probability distribution?

The experiments and results are clearly presented in Table 1. One question: Why are there "-" marks for the ASR@1 scores of the first two rows (GCG and AutoDan)? The ablation study is helpful in highlighting the usefulness of the continuous embedding space exploration.

Accept, because the paper introduces a novel and effective framework for generating adversarial suffixes, demonstrating strong empirical results and offering a scalable solution for red-teaming LLMs. The improved readability and efficiency of GASP make it a valuable contribution to the field, even though the current organization makes the paper less accessible than it could be.

Minor comments:
Appendix I, Algorithm 1 - line 946, the first mention of TargetLLM should be SuffixLLM

---

### Official Review · Reviewer_AQbh · 2025-03-02

**Rating:** 7
**Confidence:** 3

**Review:**

Summary:

This paper presents a model for efficiently generating jailbreak attacks to existing LLMs. The model, GASP, relies on adding adversarial suffixes to prompts, and is pre-trained on a set of adversarial suffixes and refined with odds-ratio preference optimization. Experiments show that GASP can compromise language models more frequently than other jailbreaking approaches and produces more readable prompts. GASP is also more efficient than some other jailbreaking frameworks.

Pros:
* The objective of creating readable prompts is interesting and an aspect that other automated frameworks for jailbreaking language models can miss. An adversarial prompt could be much more dangerous if it is difficult to detect by existing safety mechanisms.
* The paper is well-organized and concise.

Cons:
* This paper focuses on jailbreaking through adversarial suffixes, instead of jailbreak attacks in general. While GASP may be effective in compromising current language models, its lack of coverage may limit its usefulness in evaluating new defenses' ability to deal with jailbreak attacks.

---

### Decision · Program_Chairs · 2025-03-04

Accept